# MLPrune: Multi-Layer Pruning for Automated Neural Network Compression

## Abstract

Model compression can significantly reduce the computation and memory footprint of large neural networks. To achieve a good trade-off between model size and accuracy, popular compression techniques usually rely on hand-crafted heuristics and require manually setting the compression ratio of each layer. This process is typically costly and suboptimal. In this paper, we propose a Multi-Layer Pruning method (MLPrune), which is theoretically sound, and can automatically decide appropriate compression ratios for all layers. Towards this goal, we use an efficient approximation of the Hessian as our pruning criterion, based on a Kronecker-factored Approximate Curvature method. We demonstrate the effectiveness of our method on several datasets and architectures, outperforming previous state-of-the-art by a large margin. Our experiments show that we can compress AlexNet and VGG16 by **25x** without loss in accuracy on ImageNet. Furthermore, our method has much fewer hyper-parameters and requires no expert knowledge.

## 1 Introduction

Deep neural networks have proven very successful in many artificial intelligence tasks such as computer vision, natural language processing, and robotic control. In exchange for such success, modern architectures are usually composed of many stacked layers parameterized with a large number of learnable weights. This contrasts classical networks which have hundreds or even thousands times fewer parameters. As a consequence, modern architectures require considerable memory storage and intensive computation. This is problematic in applications that need to run on small embedded systems or that require low-latency to make safety-critical decisions.

Fortunately, we can compress these large networks with little to no loss of accuracy, by exploiting the fact that many redundancies exist within their parameters. A popular approach is to quantize the parameters using lower precision, thus encoding the network with fewer bits (Courbariaux et al., 2016; Gong et al., 2014; Rastegari et al., 2016; Wu et al., 2016; Zhu et al., 2016). On the other hand, *pruning* techniques (Dong et al., 2017; Guo et al., 2016; Han et al., 2015a;b; Li et al., 2016; Wen et al., 2016) aim at removing redundant connections. It has been widely exploited due to its simplicity and efficacy. In this paper, we follow this line of work and propose a new pruning approach.

Pruning algorithms differ on the pruning criteria employed. Popular pruning methods typically rely on heuristics, such as weight magnitudes. However, a small magnitude does not necessarily mean unimportance (Hassibi & Stork, 1993; LeCun et al., 1990); if the input neuron has a large expected value, a small weight could still have a large effect on its output neuron. As a consequence, magnitude-based pruning might delete important parameters, or preserve unimportant ones. Furthermore, magnitude-based pruning requires manually setting the compression ratio of each layer. These ratios are not easy to tune, as different layers (and architectures) typically have different sensitivities for compression. Directly comparing the parameters' magnitudes from all layers and pruning the smallest one would not solve the problem, since magnitudes are not calibrated across layers. Because of this, existing approaches usually involve time-consuming trial and error processes that require domain expertise.

In this paper we employ the Hessian matrix as a principled pruning criterion, as it characterizes the local curvature of the training loss. The Hessian was exploited in the past, but was either assumed to be computable (Hassibi & Stork, 1993), which is not possible in practice for modern neural networks with millions of parameters, or simplistic approximations (e.g., diagonal (LeCun et al., 1990)) were

utilized resulting in poor performance. Here we first note that the Fisher Information matrix is close to the Hessian under certain conditions. We then use a Kronecker-factored Approximate Curvature (K-FAC) (Martens & Grosse, 2015) method to efficiently estimate the Fisher matrix. In our experiments, the overhead of estimating the Hessian approximation is very small compared with the pre-train and re-train time. Therefore, our method is efficient to use, while exhibiting better compression performance compared with previous second-order pruning. Importantly, this criterion is also calibrated across layers, allowing us to directly compare parameters from different layers and set one global compression ratio for the whole network, rather than one ratio per layer. This makes our method significantly easier to use when compared to magnitude-based pruning. Our method achieves state-of-the-art results on several benchmark datasets and model architectures. Furthermore, it can be easily applied to new customized network architectures, without costly tuning many compression hyper-parameters.

## 2 BACKGROUND

In this section, we first introduce the notation used in the paper. Since our work is inspired by the Optimal Brain Surgeon (Hassibi & Stork, 1993), we also briefly review the idea and its limitations.

### 2.1 NOTATION

Suppose we have a neural network with $L$ fully-connected layers[1]. Let $p(y|x, \boldsymbol{\Theta})$ be the output distribution and $\mathcal{L}$ be the loss function. $y$ denotes the target outputs, $x$ the inputs, and $\boldsymbol{\Theta}$ the network's weights. Each layer $l$ is associated with its input $\mathbf{a}_{l-1} \in \mathbb{R}^{d_{l-1}}$, a weight matrix $\mathbf{W}_l \in \mathbb{R}^{d_{l-1} \times d_l}$, a binary mask matrix $\boldsymbol{\Gamma}_l \in \mathbb{R}^{d_{l-1} \times d_l}$ and its output $\mathbf{s}_l \in \mathbb{R}^{d_l}$. The elements $\gamma_l^{(i,j)} \in \{0, 1\}$ in the binary matrix $\boldsymbol{\Gamma}_l$ indicate whether the corresponding weight is retained or pruned away. We use subscript to denote the layer index, and superscript to denote the element index in a matrix. We also distinguish a matrix (vector) from a scalar by capital boldface[2]. The weights in a layer $l$ can be computed as $\theta_l^{(i,j)} = w_l^{(i,j)} \gamma_l^{(i,j)}$, and thus the forward-pass can be written as,

$$\mathbf{s}_l = (\mathbf{W}_l \odot \boldsymbol{\Gamma}_l) \mathbf{a}_{l-1}, \quad \mathbf{a}_l = h(\mathbf{s}_l), \tag{1}$$

where $\odot$ denotes element-wise product, and $h$ is the Relu activation in our experiments. We also use $\mathbf{W}$ to denote the concatenation of $[vec\{\mathbf{W}_1\}, vec\{\mathbf{W}_2\}, \cdots, vec\{\mathbf{W}_L\}]$, which is a column vector with the same size as the total number of parameters in this neural network, similarly for $\boldsymbol{\Gamma}$ and $\boldsymbol{\Theta}$. Index $q$ denotes the index number of a parameter in $\mathbf{W}$(or $\boldsymbol{\Gamma}$, $\boldsymbol{\Theta}$) that is associated with $w_l^{(i,j)}$(or $\gamma_l^{(i,j)}, \theta_l^{(i,j)}$). Parameters, weights and connections are used interchangeably.

### 2.2 A REVIEW ON OPTIMAL BRAIN SURGEON (OBS)

Given a neural network with parameters $\boldsymbol{\Theta}$, the local surface of the training loss can be characterized by its Taylor expansion:

$$\delta\mathcal{L} = \frac{\partial\mathcal{L}}{\partial\boldsymbol{\Theta}}\delta\boldsymbol{\Theta} + \frac{1}{2}\delta\boldsymbol{\Theta}^T \mathbf{H} \delta\boldsymbol{\Theta} + O(||\delta\boldsymbol{\Theta}||^3), \tag{2}$$

where $\delta\boldsymbol{\Theta}$ is a small change of the parameter values (e.g., setting some parameters to zero) and $\mathbf{H}$ is the Hessian matrix defined as $\partial^2\mathcal{L}/\partial\boldsymbol{\Theta}^2$. Note that at convergence the first-order term vanishes and the higher-order term can be neglected. OBS aims to find an index $q$, such that when changing $\theta_q$ from its original value to zero, the change in training loss $\delta\mathcal{L}$ is minimized:

$$\min_q \left( \min_{\delta\boldsymbol{\Theta}} \left( \frac{1}{2}\delta\boldsymbol{\Theta}^T \mathbf{H} \delta\boldsymbol{\Theta} \right) \right), \quad s.t. \quad \mathbf{e}_q^T \delta\boldsymbol{\Theta} + \delta\boldsymbol{\Theta} = 0, \tag{3}$$

where $\mathbf{e}_q^T$ is the unit vector in the weight space corresponding to $\theta_q$. The inner minimization problem can be solved by the method of Lagrangian multipliers. Hassibi & Stork (1993) applied OBS on a small network with 18,000 weights and showed good results. However, since the exact solution involves the inverse of the Hessian matrix, original OBS is intractable for modern neural networks that contain millions of parameters.

---

[1]Convolutional layers can be transformed into fully-connected layers if we vectorize filters within each channel and expand the input feature map with repeated patches

[2]with minor abuse of notation for $\mathbf{a}_{l-1}$ and $\mathbf{s}_l$

## 3 MULTI-LAYER PRUNING

Our method follows a similar pipeline when compared to conventional pruning methods. We first pre-train a neural network until converged, and then apply Multi-Layer Pruning (including surgeon) upon it. This will give us a compressed network with slightly worse performance. Finally, the network is re-trained to recover the model performance. In the following paragraph, we first introduce the problem formulation and its ideal solution in Section 3.1. We then show how to make approximations in Section 3.2.

### 3.1 MULTI-LAYER PRUNING AND SURGEON

Our objective is to prune as much as possible while maintaining good model performance, e.g. high accuracy in classification. This is similar to the objective of Minimum Description Length (MDL, (Rissanen, 1978))

$$\min_{\boldsymbol{\Theta}} \mathcal{D}(data|\boldsymbol{\Theta}) + \mathcal{D}(\boldsymbol{\Theta}). \tag{4}$$

In our case, the first term in Eq. (4) is equivalent to the training loss, and the second term is the description length denoting how many bits we need to encode the model. Here, we use the number of parameters as an indicator of such description length. We also introduce a relative importance $\lambda$, which characterizes the trade-off between sparsity and model performance, and controls the final size of a compressed model. Our objective can then be written as

$$\Psi = \min_{\mathbf{W}, \boldsymbol{\Gamma}} \left[ \mathcal{L}\left(y|x, \mathbf{W} \odot \boldsymbol{\Gamma}\right) + \lambda \sum_{l=1}^{L} \left( \sum_{(i,j)} \gamma_l^{(i,j)} \right) \right], \tag{5}$$

where $\boldsymbol{\Gamma}$ is the binary mask matrix defined in Section 2.1. This objective cannot be directly optimized with SGD as $\gamma_l^{(i,j)}$ is contrained to be binary. Instead, we consider an easier case where we first pre-train the model to the local minimum of $\mathcal{L}$, then update one $\gamma_l^{(i,j)}$ from 1 to 0 if such change will decrease $\Psi$, or equivalently, if such change will not increase $\mathcal{L}$ more than $\lambda$. Since the parameter for forward-pass is $\theta_l^{(i,j)} = \gamma_l^{(i,j)} w_l^{(i,j)}$, updating $\gamma_l^{(i,j)}$ is equivalent to prune a parameter $\theta_l^{(i,j)}$. Similar to Hassibi & Stork (1993), we also consider a surgeon operation after pruning. This will change other preserved parameters accordingly, so as to minimize the loss increment caused by the pruning. More specifically, after pruning one parameter $\theta_l^{(i,j)}$ and applying the surgeon, the increment of $\mathcal{L}$ will be,

$$\min_{\delta\boldsymbol{\Theta}} \left( \frac{1}{2} \delta\boldsymbol{\Theta}^{\mathbf{T}} \mathbf{H} \delta\boldsymbol{\Theta} \right), \quad s.t. \quad \delta\theta_l^{(i,j)} + w_l^{(i,j)} = 0. \tag{6}$$

$\delta\theta_l^{(i,j)}$ is the corresponding scalar value in $\delta\boldsymbol{\Theta}$, and $\mathbf{H}$ is the Hessian matrix of $\mathcal{L}$ w.r.t $(\mathbf{W} \odot \boldsymbol{\Gamma})$. Solving such a minimization problem with Lagrangian multipliers results in the following update

$$\Delta\mathcal{L}_q = \frac{1}{2} \frac{\left(w_l^{(i,j)}\right)^2}{[\mathbf{H}^{-1}]^{(q,q)}}, \quad \delta\boldsymbol{\Theta}^* = \left[ -\frac{w_l^{(i,j)}}{[\mathbf{H}^{-1}]^{(q,q)}} \mathbf{H}^{-1} \right]^{(.,q)}. \tag{7}$$

Here, $\delta\boldsymbol{\Theta}^*$ is the surgeon applied on the remaining $\boldsymbol{\Theta}$ (and $\mathbf{W}$ accordingly). $\Delta\mathcal{L}_q$ is the increment of loss after pruning and the surgeon operation. Again, superscript $(q, q)$ denotes the element at the $q^{th}$ row and $q^{th}$ column, $(., q)$ denotes the $q^{th}$ column in that matrix, and $q$ is the index of the parameter in $\mathbf{H}$ that associates with $w_l^{(i,j)}$. The estimation of $\mathbf{H}^{-1}$ will be introduced in section 3.2.

Theoretically, we can calculate $\Delta\mathcal{L}_q$ for each parameter and prune the one with the smallest $\Delta\mathcal{L}$, but this will become impractically time-consuming. Therefore, at each pruning step, we calculate $\Delta\mathcal{L}_q$ for each remaining parameter, and prune all parameters with $\Delta\mathcal{L}_q < \lambda$. This is fine as long as such assumption holds, that $\mathbf{H}$ does not change significantly when we prune those unimportant parameters one by one. In our experiments, we simply set $\lambda$ to be the $p^{th}$ percentile of all $\Delta\mathcal{L}_q$ from all layers. Therefore, our method can jointly determine the importance of parameters from all layers and prune multiple layers simultaneously.

After pruning and doing surgeon, we apply SGD to those retained parameters to re-train the network,

$$\mathbf{W} \leftarrow \mathbf{W} - \alpha \frac{\partial \mathcal{L}\left(\mathbf{W} \odot \boldsymbol{\Gamma}\right)}{\partial\left(\mathbf{W} \odot \boldsymbol{\Gamma}\right)} \odot \boldsymbol{\Gamma}. \tag{8}$$

## 3.2 Approximating Hessian using Fisher

Performing the pruning and surgeon operations as in Eq. (7) involves estimating and inverting the Hessian matrix, which is intractable for modern neural networks that contain millions of parameters. To efficiently calculate Eq. (7), we approximate the Hessian. Towards this goal, we first employ the Fisher Information matrix to approximate the Hessian and further use a Kronecker-factored Approximate Curvature (K-FAC) method to approximate the Fisher matrix. The first approximation comes from the fact that if the training objective is the negative log-likelihood, the Hessian matrix consists of expected second-order derivatives under the data distribution, while the Fisher matrix consists of expected second-order derivatives under the model distribution. Since modern neural networks usually have strong model capacities, we expect those two distributions are close for a well-trained model. The second approximation is demonstrated to be reasonable in optimization tasks (Grosse & Martens, 2016; Martens & Grosse, 2015), and will further help us to calculate Eq. (7) efficiently.

The K-FAC method approximates the Fisher matrix as a block-diagonal matrix and estimates each block by the Kronecker-product of two much smaller matrices, which are the second-order statistics of inputs and derivatives. Given a neural network with stacked fully-connected layers[3], the Fisher Information matrix is defined as

$$\mathbf{F} = \mathbb{E}\left[\left(\nabla_{\mathbf{W}}\mathcal{L}\right)\left(\nabla_{\mathbf{W}}\mathcal{L}\right)^T\right]. \tag{9}$$

Unless specified otherwise, the expectation is taken with respect to the model distribution. We can re-write the Fisher in a block-wise manner by partitioning rows and columns of $\mathbf{F}$ if they correspond to parameters within the same layer. The $(i, j)$ block is then[4]

$$\mathbf{F}_{ij} = \mathbb{E}\left[vec\left\{\nabla_{\mathbf{W}_i}\mathcal{L}\right\} vec\left\{\nabla_{\mathbf{W}_j}\mathcal{L}\right\}^T\right], \tag{10}$$

which is the covariance of the gradients from $i$ and $j$ layers[5]. As noted by Martens & Grosse (2015), the elements of the covariance of derivatives from different layers are generally smaller than the ones within the same layer. Therefore, $\mathbf{F}$ can be approximated by a block-diagonal matrix where the $l^{th}$ block is simply $\mathbf{F}_{ll}$.

Using back-propagation, the gradients of layer $l$ can be calculated as $\nabla_{\mathbf{W}_l}\mathcal{L} = \left(\nabla_{\mathbf{s}_l}\mathcal{L}\right)\left(\mathbf{a}_{l-1}\right)^T$. The $l^{th}$ diagonal block $\mathbf{F}_{ll}$ can then be written as

$$\mathbf{F}_{ll} = \mathbb{E}\left[vec\left\{\left(\nabla_{\mathbf{s}_l}\mathcal{L}\right)\left(\mathbf{a}_{l-1}\right)^T\right\} vec\left\{\left(\nabla_{\mathbf{s}_l}\mathcal{L}\right)\left(\mathbf{a}_{l-1}\right)^T\right\}^T\right] = \mathbb{E}\left[\mathbf{a}_{l-1}\mathbf{a}_{l-1}^T \otimes \left(\nabla_{\mathbf{s}_l}\mathcal{L}\right)\left(\nabla_{\mathbf{s}_l}\mathcal{L}\right)^T\right] \tag{11}$$

$$\approx \mathbb{E}\left[\mathbf{a}_{l-1}\mathbf{a}_{l-1}^T\right] \otimes \mathbb{E}\left[\left(\nabla_{\mathbf{s}_l}\mathcal{L}\right)\left(\nabla_{\mathbf{s}_l}\mathcal{L}\right)^T\right], \tag{12}$$

where $\otimes$ denotes Kronecker-product. The second equality comes from the property of the Kronecker-product[6]. The last approximation is supposed to maintain the "coarse structure" of the Fisher matrix (Martens & Grosse, 2015), and such "coarse structure" is sufficient for our method to achieve a good compression ratio as empirically demonstrated in our experiments. Therefore, $\mathbf{F}$ (and thus $\mathbf{H}$) can be approximated by several much smaller matrices

$$\mathbf{A}_{l-1} = \mathbb{E}\left[\mathbf{a}_{l-1}\mathbf{a}_{l-1}^T\right], \quad \mathbf{DS}_l = \mathbb{E}\left[\left(\nabla_{\mathbf{s}_l}\mathcal{L}\right)\left(\nabla_{\mathbf{s}_l}\mathcal{L}\right)^T\right], \quad \mathbf{F}_{ll} = \mathbf{A}_{l-1} \otimes \mathbf{DS}_l. \tag{13}$$

For a typical modern neural network such as AlexNet, the original Hessian matrix is a $61\text{M} \times 61\text{M}$ matrix, while $\mathbf{A}_{l-1}$ and $\mathbf{DS}_l$ of the largest fully-connected layer have sizes of only $9126 \times 9126$ and $4096 \times 4096$ respectively. We use exponential moving average to estimate the expectation in $\mathbf{A}_{l-1}$ and $\mathbf{DS}_l$, with a decay factor of 0.95 and a horizon of 1000 steps in all experiments. This adds only a small overhead to the normal forward-backward pass. Furthermore, the inverse $\mathbf{F}^{-1}$ and matrix-vector product $\mathbf{F}^{-1}\mathbf{h}$ can be efficiently calculated by leveraging the block diagnoal structure and property of Kronecker-product[7].

---

[3]Details for convolutional layers are discussed in Grosse & Martens (2016).

[4]Note that $vec\{\}$ denotes vectorization, as $\mathbf{W}_i$ is a matrix, while previously $\mathbf{W}$ is a vector. See section2.1

[5]$\mathbb{E}\left[\nabla_{\mathbf{W}}\mathcal{L}\right] = \mathbf{0}$, since the expectation is taken over the model distribution

[6]$vec\left\{\mathbf{u}\mathbf{v}^T\right\} = \mathbf{v} \otimes \mathbf{u}, \quad (\mathbf{A} \otimes \mathbf{B})(\mathbf{C} \otimes \mathbf{D}) = (\mathbf{A} \otimes \mathbf{C})(\mathbf{B} \otimes \mathbf{D})$.

[7]$(\mathbf{A} \otimes \mathbf{B})^{-1} = \mathbf{A}^{-1} \otimes \mathbf{B}^{-1}, \quad (\mathbf{A} \otimes \mathbf{B})vec\{\mathbf{X}\} = vec\{\mathbf{B}\mathbf{X}\mathbf{A}^T\}$

---

**Algorithm 1** Multi-Layer Pruning

---

**Initialization:** $\mathbf{W}, \mathbf{\Gamma}$, Pruning fraction $p$.
**Pre-training Stage:**
 1: Pre-train the network
**Pruning Stage:**
 2: **for** $t = 0, \cdots, 1000$ **do**
 3:     Update $\mathbf{A}_{l-1} \leftarrow 0.95 \times \mathbf{A}_{l-1} + 0.05 \times \mathbf{a}_{l-1}\mathbf{a}_{l-1}^T$
 4:     Update $\mathbf{DS}_l \leftarrow 0.95 \times \mathbf{DS}_l + 0.05 \times (\nabla_{\mathbf{s}_l}\mathcal{L})(\nabla_{\mathbf{s}_l}\mathcal{L})^T$
 5: **end for**
 6: Compute Fisher matrix $\mathbf{F}$ by Eq. (13).
 7: Compute importance measure $\Delta\mathcal{L}_q$ for each parameter by Eq. (7).
 8: Normalize $\Delta\mathcal{L}_q$ within each layer by Eq. (14).
 9: Compute $p^{th}$ percentile of $\Delta\tilde{\mathcal{L}}_q$ from all layers as $\lambda$.
10: Update mask $\gamma_l^{(i,j)}$ to 0 if its corresponding $\Delta\tilde{\mathcal{L}}_q$ is smaller than $\lambda$.
11: Compute $\delta\mathbf{\Theta}^*$ by Eq. (7) and update $\mathbf{\Theta}, \mathbf{W} \leftarrow \mathbf{\Theta} + \delta\mathbf{\Theta}^*$.
12: Re-train the network by Eq. (8).

---

However, such a block diagonal approximation decorrelates the gradients from different layers, and might lead to different scales for different layers. This will become problematic when we jointly prune all layers, i.e. sorting $\Delta\mathcal{L}_q$ from all layers. We empirically find that $\Delta\mathcal{L}_q$ from different layers will sometimes differ by several orders of magnitudes, which makes it impossible to perform the multi-layer pruning. Therefore, we propose a simple yet effective modification for this problem. After calculating $\Delta\mathcal{L}_q$, we normalize it within the same layer:

$$\Delta\tilde{\mathcal{L}}_q = \frac{\Delta\mathcal{L}_q}{\sum_{q \in \text{layer } l} \Delta\mathcal{L}_q} \tag{14}$$

Then, we use $\Delta\tilde{\mathcal{L}}_q$ as an importance measure to prune parameters. Surgeon and other parts remain unchanged. We show our algorithm as Algorithm 1. During pruning, we only need to set an overall compression ratio, and the MLPrune method will automatically decide the compression ratio for each layer. This free us from tuning hyper-parameters, and thus it is easier and more efficient to apply.

# 4 RELATED WORK

Model compression task aims to compress a large model into a smaller one while maintaining good performance. There are several popular families of approaches, including pruning, quantization, and low-rank approximation. Quantization (Courbariaux et al., 2016; Gong et al., 2014; Rastegari et al., 2016; Wu et al., 2016; Zhu et al., 2016) aims to use fewer bits to encode each parameter, e.g. binary neural network. Low-rank approximation (Denton et al., 2014; Jaderberg et al., 2014; Lebedev et al., 2014; Novikov et al., 2015) approximates layer weights by low-rank representations, and thus saves the storage and makes speedup. Pruning, being one of the most popular methods due to its simplicity and effectiveness, aims to delete unimportant parameters from a large network. These techniques could be further integrated together and result in better compression ratio (Han et al., 2015a).

The pruning operation involves finding and deleting unimportant parameters, and thus needs a criterion for deciding importance. Magnitude-based methods (Guo et al., 2016; Han et al., 2015b; Li et al., 2016; Wen et al., 2016) use the absolute value of a parameter as an indicator, assuming smaller weights generally have smaller importance upon a network's output, but there is no theoretical evidence to support this criterion. On the other hand, some other work (Hassibi & Stork, 1993; LeCun et al., 1990) use the second-order Hessian matrix to calculate the importance of each parameter, and show significant improvements when applied to toy neural networks. However, these methods cannot be easily adapted to modern neural networks due to the large size of the Hessian matrix. Recently, Dong et al. (2017) use the layer-wise reconstruction as an indicator, which provides a theoretical upper bound for the pruning error, while avoiding the intractable large Hessian matrix. However, all these methods require specifying one compression ratio for each layer, and searching such ratios are time-consuming for very deep neural networks.

Table 1: Compression Ratios (CR) of Different Architectures

| Method | Architecture | Original Error | Final Error | Δ Error | CR |
|---|---|---|---|---|---|
| Random | LeNet-300-100 | 1.76% | 2.25% | 0.49% | 8% |
| OBD | LeNet-300-100 | 1.76% | 1.96% | 0.20% | 8% |
| LWC | LeNet-300-100 | 1.64% | 1.59% | -0.05% | 8% |
| DNS | LeNet-300-100 | 2.28% | 1.99% | **-0.29%** | 1.8% |
| L-OBS | LeNet-300-100 | 1.76% | 1.96% | 0.20% | 1.5% |
| MLPrune(Ours) | LeNet-300-100 | 1.86% | 1.94% | 0.08% | **1.3%** |
| OBD | LeNet-5 | 1.27% | 2.65% | 1.38% | 8% |
| LWC | LeNet-5 | 0.80% | 0.77% | **-0.03%** | 8% |
| DNS | LeNet-5 | 0.91% | 0.91% | 0.00% | 0.9% |
| L-OBS | LeNet-5 | 1.27% | 1.66% | 0.39% | 0.9% |
| MLPrune(Ours) | LeNet-5 | 0.85% | 0.89% | 0.04% | **0.5%** |
| LWC | Cifar-Net | 18.57% | 19.36% | 0.79% | 9% |
| L-OBS | Cifar-Net | 18.57% | 18.76% | 0.19% | 9% |
| MLPrune(Ours) | Cifar-Net | 18.43% | 18.60% | **0.17%** | **6.4%** |
| DNS | AlexNet | 43.42% | 43.09% | **-0.33%** | 5.7% |
| LWC | AlexNet | 42.78% | 42.77% | -0.01% | 11% |
| L-OBS | AlexNet | 43.30% | 43.11% | -0.19% | 11% |
| MLPrune(Ours) | AlexNet | 43.17% | 43.14% | -0.03% | **4.0%** |
| DNS[8] | VGG16 | 31.66% | 63.38% | 31.72% | 7.5% |
| LWC | VGG16 | 31.50% | 31.44% | -0.06% | 7.5% |
| L-OBS | VGG16 | 31.66% | 32.02% | 0.36% | 7.5% |
| MLPrune(Ours) | VGG16 | 30.95% | 30.78% | **-0.17%** | **4.0%** |

In our work, we utilize K-FAC to approximate the Fisher matrix, which in turn approximates the exact Hessian matrix. K-FAC(Grosse & Martens, 2016; Martens & Grosse, 2015) provides an efficient way to estimate and invert an approximation of the Fisher matrix of a neural network. It first approximates the Fisher by a block diagonal matrix, and then decomposes each block by two much smaller matrix via Kronecker-product. Such approximation and its variants(Ba et al., 2016; Wu et al., 2017; Zhang et al., 2017) have shown success in the field of optimization.

## 5 EXPERIMENTS

Following previous work, we use compression ratio as our evaluation metric, which is defined as the ratio of the number of parameters that remaining active after pruning to the number of original parameters. To show the generality of our method, we perform experiments using three different datasets and five different deep network architectures, from shallow to deep. This includes LeNet-300-100 and LeNet-5 (LeCun et al., 1998) on MNIST, CifarNet (Krizhevsky & Hinton, 2009) on Cifar-10, and AlexNet (Krizhevsky et al., 2012) and VGG16(Simonyan & Zisserman, 2014) on Imagenet ILSVRC-2012. The aforementioned architectures contain both fully-connected layers and convolution layers, and have different sizes from 267K to 138M parameters. We compare our results with some strong pruning baselines, including: 1) Randomly Pruning (Dong et al., 2017), 2) OBD (LeCun et al., 1990), 3) LWC (Han et al., 2015b), 4) DNS (Guo et al., 2016), 5) L-OBS (Dong et al., 2017). Following these work, we also apply pruning in an iterative manner and report the results in Table 1. Due to minor differences in implementation, different papers report different performance of pre-trained models. Therefore, we also report the original error and Δerror before and after pruning for each method. More implementation details could be found in Appendix A.

### 5.1 DOES MLPRUNE ACHIEVES BETTER COMPRESSION RESULTS?

**MNIST:** We first conduct experiments on the MNIST dataset with LeNet-300-100 and LeNet-5. LeNet-300-100 has 2 fully-connected layers, with 300 and 100 hidden units respectively. LeNet-5 is a CNN with 2 convolutional layers, followed by 2 fully-connected layers. Table 1 shows that we can compress LeNet-300-100 to 1.3% of original size with almost no loss in performance. Similarly for LeNet-5, we can compress to 0.5%, which is much smaller than previous best result.

---

[8]Original DNS paper doesn't provide result on VGG16. Therefore we adopt numbers from the re-implementation in Dong et al. (2017)

**Cifar-10:** We conduct experiments on Cifar-10 image classification benchmark with Cifar-Net architecture. Cifar-Net is a variant of AlexNet, containing 3 convolutional layers and 2 fully-connected layers. Following previous work (Dong et al., 2017), we first pre-train the network to achieve 18.43% error rate on the testing set. We then iteratively prune the network to $[50\%, 25\%, 12.5\%, 10\%, 8\%, 6.4\%]$ of its original size. This gives us a better compression ratio than our baselines while maintaining similar performance.

**ImageNet:** To demonstrate our method's effectiveness on larger models and datasets, we prune AlexNet on ImageNet ILSVRC-2012. AlexNet has 5 convolutional layers, 3 fully-connected layers, and 61M parameters in total. We first pre-train the network, and achieve similar top-1 accuracy with other implementations. We then iteratively prune the network to $[50\%, 25\%, 12.5\%, 6.25\%, 5\%, 4\%]$ of its original size. To the best of our knowledge, this is the best pruning result so far. Also notice that Han et al. (2015b) and Guo et al. (2016) find out it is necessary to fix the convolutional layers when pruning and retraining the fully-connected layer (and vice versa), otherwise the model cannot recover the pruning error. However, we don't observe such difficulty in our experiments, as we simply prune and retrain all layers simultaneously. This also indicates that our pruning operation only brings small error and still preserves the model capacity.

We further apply our method to a modern neural network architecture, VGG16. VGG16 is a deep network with 13 convolutional layers, 3 fully-connected layers and 138M parameters in total. Due to its large number of layers, the search space of per-layer compression ratio is exponentially larger than that of AlexNet. Therefore, conventional methods would have difficulties to tune and set the compression ratio for each of those 16 layers. On the contrary, our method can be used in a *push-button-to-start* manner, and automatically finds out appropriate ratios. We use the same pruning set-up as in AlexNet experiment, and achieve the state-of-the-art result as shown in Table 1.

## 5.2 DOES MLPRUNE FIND REASONABLE PER-LAYER COMPRESSION RATIOS?

We compare the compression ratios, layer by layer, between our method and the baselines, as shown in Table 2. The compression ratios of Han et al. (2015b) and Guo et al. (2016) are manually tuned to achieve good results. In contrast, our method automatically determines the compression ratio for each layer during the pruning process. It involves no time-consuming procedure for tuning hyper-parameters yet achieves better results. We also notice that our compression ratios are not the same as previous manually-tuned ratios, but share some similarities; layers with smaller compression ratios in our method also have smaller ratios in the baselines. Also, fully-connected layers are generally be pruned more severe than convolutional layers, which is in accord with the observation that fully-connected layers usually have more redundancies. These suggest that our method can find reasonable per-layer compression ratios according to the sensitivities of each layer. Detailed compression ratios for other architectures could be found in Appendix B.

Table 2: AlexNet Per-layer Compression Ratios

| Architecture | Layer | Parameters | Han et al. (2015b) | Guo et al. (2016) | MLPrune (Ours) |
|---|---|---|---|---|---|
| | conv1 | 35K | 84% | 53.8% | 67.2% |
| | conv2 | 307K | 38% | 40.6% | 37.8% |
| | conv3 | 885K | 35% | 29.0% | 27.7% |
| | conv4 | 663K | 37% | 32.3% | 33.2% |
| AlexNet | conv5 | 442K | 37% | 32.5% | 38.6% |
| | fc1 | 38M | 9% | 3.7% | 1.5% |
| | fc2 | 17M | 9% | 6.6% | 3.2% |
| | fc3 | 4M | 25% | 4.6% | 13.7% |
| | total | 61M | 11% | 5.7% | **4%** |

## 5.3 DOES MLPRUNE PRUNE SIMILAR PARAMETERS AS MAGNITUDE-BASED PRUNING?

In section 5.2, we've seen MLPrune can automatically adjust how much to prune from different layers. In this section, we'll investigate what parameters does MLPrune prune within one layer. Since both MLPrune and magnitude-based pruning can achieve good compression results, it's intriguing

to explore if MLPrune prune similar parameters as magnitude-based pruning, or they prune totally different parameters.

Figure 1 shows the distribution of parameters' magnitudes, coming from the first fully-connected layer of AlexNet. Before pruning, weight distribution is peaked at 0, and drops quickly as the absolute value increasing. This is very much like a gaussian distribution as what the weights were initialized from. Figure 1b shows the distribution after pruning using our method. It is obvious to see that parameters with magnitudes close to 0 (center region) are pruned away, indicating that the parameters regarded as having small impacts upon the loss by our method also usually have small magnitudes.

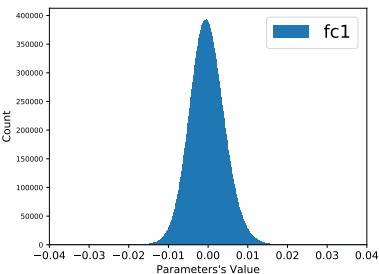
(a) Weight distribution before pruning

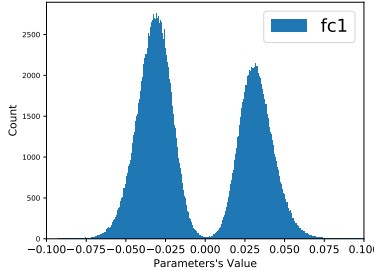
(b) Weight distribution after pruning

Figure 1: Weight distribution of the first FC layer of AlexNet before and after pruning.

We further explore the correlation between a parameter's magnitude and its importance measured by our method. From figure 2, we can observe that: 1) A parameter's absolute value indeed has a strong correlation with its importance computed by MLPrune. This explains why magnitude-based methods also achieve fairly good performance. Despite this strong correlation, directly compare the magnitudes from all layer will give different pruning result than our method: setting the pruning threshold as the median of parameters' magnitudes in Figure 2a will prune 14% of conv5, while the median of our importance will prune 7%. This explains why magnitude-based pruning cannot do multi-layer pruning. 2) FC layer initially has smaller importance(larger redundancies) than convolutional layer measured by MLPrune, and thus are pruned more severely. As more and more parameters are pruned away(from Figure 2a to Figure 2b), the importance of FC layer becomes closer to that of convolutional layer, and both layers will be pruned with similar ratios. This shows that our method can dynamically adjust the compression ratio of each layer, as the pruning going on.

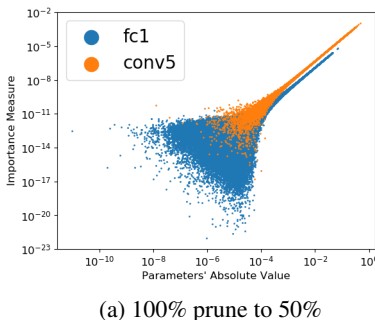
(a) 100% prune to 50%

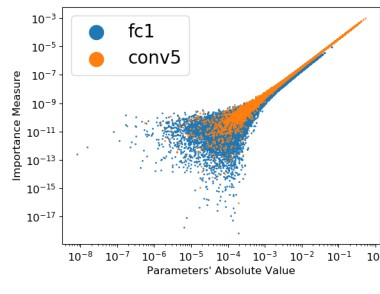
(b) 25% prune to 12.5%

Figure 2: Correlation between parameters' absolute values and importances.

## 6 CONCLUSION

In this paper we have proposed an automated Multi-Layer Pruning (MLPrune) method to compress deep neural networks. Our approach exploits an efficient approximation of the Hessian based on the K-FAC method as our pruning criterion. It has much fewer hyper-parameters and can be used in an automated manner without costly test and trial processes. We have demonstrated the effectiveness of our method on several datasets and architectures and achieved state-of-the-art results. In the future, we plan to extend our method to prune coarser components such as filters, rather than individual weights, so as to speed up the network and save memory furthermore.

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

# A   PRUNING SETUP

In this section, we provide the implementation details for experiments of AlexNet and VGG16.

Table 3: Retraining Setup

| Architecture | Pruning Phase | Init LR | Decay Epochs | Decay Rate | Dropout | Weight Decay | Stopping Epoch |
|---|---|---|---|---|---|---|---|
| | pretrain | 0.1 | 40 | 0.1 | 0.5 | 5e-4 | 120 |
| | 100% to 50% | - | - | - | - | - | - |
| | 50% to 25% | 0.1 | 40 | 0.1 | 0.5 | 2e-4 | 120 |
| AlexNet | 25% to 12.5% | 0.1 | 40 | 0.1 | 0.4 | 2e-4 | 120 |
| | 12.5% to 6.25% | 0.1 | 40 | 0.1 | 0.4 | 2e-4 | 120 |
| | 6.25% to 5% | 0.1 | 40 | 0.1 | 0.4 | 2e-4 | 120 |
| | 5% to 4% | 0.1 | 40 | 0.1 | 0.3 | 2e-4 | 120 |
| | pretrain | 0.05 | 20 | 0.1 | 0.5 | 5e-4 | 60 |
| | 100% to 50% | - | - | - | - | - | - |
| | 50% to 25% | 0.05 | 20 | 0.1 | 0.5 | 2e-4 | 60 |
| VGG16 | 25% to 12.5% | 0.05 | 20 | 0.1 | 0.5 | 2e-4 | 60 |
| | 12.5% to 6.25% | 0.05 | 20 | 0.1 | 0.5 | 2e-4 | 60 |
| | 6.25% to 5% | 0.05 | 20 | 0.1 | 0.5 | 2e-4 | 60 |
| | 5% to 4% | 0.05 | 20 | 0.1 | 0.4 | 2e-4 | 60 |

The AlexNet is first pre-trained using similar settings as in Caffe model zoo. A training image is first rescale so that the shorter edge is 256 with aspect ratio unchanged. Then a 227x227 image is randomly croped, followed by randomly flip horizontally or not. We do not apply color augmentation as in original paper (Krizhevsky et al., 2012). During testing, a center cropped image is fed into the network. SGD with momentum is applied for training. The batch-size is 256 and momentum is 0.9 for pre-training and all retraining stages.

Table 3 shows the hyper-parameters used for retraining stages (We do not apply retraining when prune from 100% to 50% since the model performance doesn't decrease before and after pruning). The learning rate schedule is exactly the same as pre-training, and doesn't require any tuning. The weight decay is slightly decreased to regularize the model less, since a pruned model has smaller representation capacity. The only hyper-parameters need to tune during the pruning is the dropout rate. As more and more parameters are pruned away, the model will become under-fit rather than over-fit. Therefore, reduce the dropout rate accordingly will be helpful to get better model performance. (Han et al., 2015b) propose a heuristic formula to determine the appropriate dropout rate for magnitude-based pruning methods. However, applying their dropout rate on our model lead to severe over-fitting. For example, when compressed to $12.5\%$, their formula suggest dropout rate of $0.18\%$, while our model adopts $0.4\%$. One possible explanation is that magnitude-based pruning methods sometimes mistakenly prune away important connections. This will significantly harm the model capacity and need much smaller dropout rate.

We adopt similar strategy for VGG16, namely using the same learning rate schedule, decrease and fix the weigh decay, and fine tune the dropout rate during pruning. The batch-size is $128$ and momentum is $0.9$ for all experiments. Again, we achieve state-of-the-art pruning result. Also notice that the dropout rate for VGG16 is larger than that of AlexNet, which might due to the fact that VGG16 has larger fully-connected layer, and thus pruning a fraction of it won't affect the model capacity that much.

## B  PER-LAYER COMPRESSION RATIOS

We compare the compression ratio, layer by layer, between our method and the baselines, as shown in Table 4. The compression ratios of Han et al. (2015b) and Guo et al. (2016) are manually tuned to achieve good results. In contrast, our method automatically determines the compression ratio for each layer during the pruning process. It involves no time-consuming procedure for tuning hyper-parameters yet achieves better results.

Table 4: Per-layer Compression Ratios

| Architecture | Layer | Parameters | Han et al. (2015b) | Guo et al. (2016) | MLPrune (Ours) |
|---|---|---|---|---|---|
| LeNet-300-100 | fc1 | 235K | 8% | 1.8% | 0.73% |
| | fc2 | 30K | 9% | 1.8% | 4.74% |
| | fc3 | 1K | 26% | 5.5 % | 39.4% |
| | total | 267K | 8% | 1.8% | **1.3%** |
| LeNet-5 | conv1 | 0.5K | 66% | 14.2% | 41.2% |
| | conv2 | 25K | 12% | 3.1% | 3.1% |
| | fc1 | 400K | 8% | 0.7% | 0.2% |
| | fc2 | 5K | 19% | 4.3% | 8.9% |
| | total | 431K | 8% | 0.9% | **0.5%** |
| AlexNet | conv1 | 35K | 84% | 53.8% | 67.2% |
| | conv2 | 307K | 38% | 40.6% | 37.8% |
| | conv3 | 885K | 35% | 29.0% | 27.7% |
| | conv4 | 663K | 37% | 32.3% | 33.2% |
| | conv5 | 442K | 37% | 32.5% | 38.6% |
| | fc1 | 38M | 9% | 3.7% | 1.5% |
| | fc2 | 17M | 9% | 6.6% | 3.2% |
| | fc3 | 4M | 25% | 4.6% | 13.7% |
| | total | 61M | 11% | 5.7% | **4%** |
| VGG16 | conv1_1 | 2K | 58% | - | 92.1% |
| | conv1_2 | 37K | 22% | - | 66.4% |
| | conv2_1 | 74K | 34% | - | 55.7% |
| | conv2_2 | 148K | 36% | - | 46.9% |
| | conv3_1 | 295K | 53% | - | 38.2% |
| | conv3_2 | 590K | 24% | - | 30.8% |
| | conv3_3 | 590K | 42% | - | 30.5% |
| | conv4_1 | 1M | 32% | - | 20.6% |
| | conv4_2 | 2M | 27% | - | 12.9% |
| | conv4_3 | 2M | 34% | - | 12.8% |
| | conv5_1 | 2M | 35% | - | 13.1% |
| | conv5_2 | 2M | 29% | - | 13.9% |
| | conv5_3 | 2M | 36% | - | 13.3% |
| | fc1 | 103M | 4% | - | 0.3% |
| | fc2 | 17M | 4% | - | 1.9% |
| | fc3 | 4M | 23% | - | 9.2% |
| | total | 138M | 7.5% | - | **4%** |

