# OpenReview forum: "MLPrune: Multi-Layer Pruning for Automated Neural Network Compression"
_ICLR.cc/2019/Conference_

### Official Review · AnonReviewer2 · 2018-10-30
**Hyper-parameter-free approach, but limited novelty**

**Rating:** 4
**Confidence:** 4

**Review:**

This paper introduces an approach to pruning the parameters of a trained neural network. The idea is inspired by the Optimal Brain Surgeon method, which relies on second derivatives of the loss w.r.t. the network parameters. Here, the corresponding Hessian matrix is approximated using the Fisher information to make the algorithm scalable to very deep networks.

Strengths:
- The method does not require hyper-parameter tuning.
- The results show the good behavior of the approach.

Weaknesses:

Novelty:
- In essence, this method relies on the work of Marten & Grosse to approximate the Hessian matrix used in the Optimal Brain Surgeon strategy. This is fine, but not of great novelty.

Method:
- It is not clear to me why the notion of binary parameters gamma is necessary. Instead of varying the gammas from 1 to 0, why not directly zero out the corresponding network parameters w?
- In essence, the objective function in Eq. 5 adds an L_1 penalty on the gamma parameters, which would be related to an L_1 penalty on the ws. Note that this strategy has been employed in the past, e.g., Collins & Kohli, 2014, "Memory Bounded Deep Convolutional Networks".
- It is not clear to me how zeroing out individual parameters will truly allows one to reduce the model afterwards. In fact, one would rather want to remove entire rows or columns of the matrix W_l, which would truly correspond to a smaller model. This was what was proposed by Wen et al., NIPS 2016 and Alvarez & Salzmann, NIPS 2016, "Learning the Number of Neurons...".
- In the past, when dealing with the Hessian matrix, people have used the so-called Pearlmutter trick (Pearlmutter, Neural Computation 2014, "Fast exact multiplication by the Hessian". In fact, in this paper, the author mentions the application to the Optimal Brain Surgeon strategy. Is there a benefit of the proposed approach over this alternative strategy?

Experiments:
- While the reported compression rates are good, it is not clear to me what they mean in practice, because the proposed algorithm zeroes out individual parameters in the matrix W_l of each layer.  This does not guarantee entire channels to be removed. As such, I would not know how to make the model actually smaller in practice. It would seem relevant to show the true gains in memory usage and in inference speed (both measured on the computer, not theoretically).

Summary:
I do appreciate the fact that the proposed method does not require hyper-parameters and that it seems to yield higher compression rates than other pruning strategies that act on individual parameters. However, novelty of the approach is limited, and I am not convinced of its actual benefits in practice.

---

> ### Author Response · Authors · 2018-11-27
> **Thanks for your comments.**
>
> Thanks for your comments and references. Hopefully our responses can answer some of your questions.
>
> -    It is not clear to me...
> Answer: We agree there are several alternatives to describe the pruning operation. We introduce this \gamma notion just for better explanation. We’ll try to simply our notations in the revised version.
>
> -    In essence...
> Answer: Thanks for your reference, we’ll add this reference in our revised version. We agree that Eq(5) introduce a L_0 penalty on the weights (In essence most pruning parameter methods have this similar objective: maintaining data loss while having less parameters), but we think our method is different from the mentioned one. From our understanding, that work adds a l1 regularizer to encourage sparsity, and then projects weights to a l0 ball which is similar to prune weights with lower magnitude. However, our method use second-order pruning criterion to select un-important weights, rather than using the magnitudes.
>
> - It is not clear to me...
> Answer: We agree that zeroing out individual parameters doesn’t directly facilitate inference speed on a standard hardware, but smaller parameter means potentially smaller size of the model, which will help mobile devices and on-chip inference(such as FPGA).
>
> - In the past...
> Answer: Thanks for the reference. To the best of our knowledge, it's still difficult to apply such technique doing the second-order pruning. For example, to estimate the importance of different weights, we need to know all the diagonal elements of the inverse of Hessian (Eq(7)). According to this technique, for each diagonal element, we need to solve argmin_x(Hx - (0, 0, ..., 0, 1, 0, …, 0)^T) using this technique and the conjugate gradient. Therefore, for a modern network, we need to apply this technique and conjugate gradient for millions of times in order to see which parameters are un-important (solving Eq(7)), and thus being impractical. We’ll add this comparison in our revised version if we later find more related work on similar topics.
>
> Experiments:
> -    While the reported compression rates are good...
> Answer: We understand the reviewer’s concern about the practical benefit of such method. However, as we mentioned before, pruning individual weights can help on-chip inference (FPGA) or mobile device where memory is an issue and customization can be applied to facilitate inference speed. Pruning individual weights can also help make model smaller as shown in [4]. Lastly, our contribution mainly focus on providing a hyper-parameter free manner to prune the network to get a smaller size, not claiming to have faster inference speed. Therefore, we think gaining speed is beyond the scope of this work.
> [4] Deep Compression: Compressing Deep Neural Networks with Pruning, Trained Quantization and Huffman Coding

---

### Official Review · AnonReviewer3 · 2018-10-31
**Marginally above acceptance threshold**

**Rating:** 6
**Confidence:** 4

**Review:**

The paper proposes a multi-layer pruning method called MLPrune for neural networks, which can automatically decide appropriate compression ratios for all the layers. It firstly pre-trains a network. Then it utilizes K-FAC to approximate the Fisher matrix, which in turn approximates the exact Hessian matrix of training loss w.r.t model weights. The approximated Hessian matrix is then used to estimate the increment of loss after pruning a connection. The connections from all layers with the smallest loss increments are pruned and the network is re-trained to the final model.

Strength:
1. The paper is well-written and clear.
2. The method is theoretically sound and outperforms state-of-the-art by a large margin in terms of compression ratio.
3. The analysis of the pruning is interesting.

Weakness:
*Method complexity and efficiency are missing, either theoretically or empirically.*
The main contribution claimed in the paper is that they avoid the time-consuming search for the compression ratio for each layers. However, there are no evidences that the proposed method can save time. As the authors mention, AlexNet contains roughly 61M parameters. On the other hand, the two matrices A_{l-1} and DS_l needed in the method for a fully-connected layer already have size 81M and 16M respectively. Is this only a minor overhead, especially when the model goes deeper?

Overall, it is a good paper. I am inclined to accept, and I hope that the authors can show the complexity and efficiency of their method.

---

> ### Author Response · Authors · 2018-11-27
> **Method Complexity and Time**
>
> Thanks for your comments. Here we provide a brief complexity and time evaluation of our method, using AlexNet as an example.
>
>     Complexity: Our method computes the Hessian in a block-wise manner, and the size of each block is determined by the size of that layer. The largest fully-connected layer in AlexNet is fc1, which is a 9216 x 4096 matrix. As a result, a_{l-1} in Eq(13) is a vector of size 9216, and \nabla_{s_l}L in Eq(13) is a vector of size 4096. Thus, A_{l-1} for this layer is a matrix of size 9216 x 9216, and DS_l has size 4096 x 4096. It is not difficult to invert two matrices with such size with standard hardwares (O(9216^3)).
>
>     Time: Each pruning operation is followed by a re-training procedure (just as other popular pruning methods). Using AlexNet as an example, the re-training procedure run 120 epochs over the ImageNet dataset, which typically involves 1-2 days on 4 Nvidia 1080 Ti and a 32 core CPU, while the pruning operation is around 70s on the same hardware. Therefore, we think our pruning method only brings negligible overhead.
>
>     In general, we do not claim that our method converge faster than other pruning methods. But we think our method can automatically determine compression ratios for every layers, and thus avoiding lots of tuning and trials for manually searching those ratios, which make our method easier and faster to use in practice.

---

> > ### Comment · AnonReviewer3 · 2018-11-29
> > **Concerns Now Addressed**
> >
> > Thank you for clarifying the complexity. Please also include it in the paper formally.

---

### Official Review · AnonReviewer1 · 2018-11-04
**Second order method for pruning multiple layers**

**Rating:** 5
**Confidence:** 5

**Review:**

This paper proposes a multi-layer pruning technique based on the Hessian. The main claims are performing better than other second order pruning methods and be principled.


Main concerns / comments are:
-	Part of the novelty relays on computing the Hessian, and the algorithm goes for very large networks (parameter wise), why? Modern networks do have much fewer parameters and do perform better. How does it behave on those? Would be interesting to see impact on modern networks (e.g., ResNet).


-	Paper claims to be principled (as many others) and being able to address multiple layers at the same time. I do believe first order methods do that as well. Why not compared to them?
-	Paper claims little overhead (compared to training and re-training). There is not much on that. Also, following the pipeline [train-prune-retrain] can be substituted by pruning while training with little overhead as in recent papers: (such as Learning with structured sparsity or Learning the number of neurons in DNN both at NIPS2016 or encouraging low-rank at compression aware training of DNN, nips 2017). Compared to those newer methods, this proposal has a drop-in accuracy while those do not. Would be nice to have a discussion related to that. Would be possible to include this into the original training process?
-	Experiments are shown in small datasets and non-current networks with millions of parameters which do not reflect current state of the art. I would be interested to see limitations in networks not having fully connected layers with the large majority of (redundant) parameters.
-	Compute time is not provided. Please comment on that
-	I am not sure if I understand the statement on 'pruning methods can not handle multiple layers'. To the best of my understanding, current pruning methods as those mentioned above do

-	Different to others, the proposed method, given a desired compression ratio can adjust the relevance of each layer. That is interesting, however, what is the motivation behind? Would be interesting to be able to control specifically each layer to make sure, for instance, the latency of each layer is maintained.


-	I am confused with \lambda, how does this go from percentage to per parameter? Is that guaranteed?

---

> ### Author Response · Authors · 2018-11-27
> **New experimental results of ResNet50 and other responses.**
>
> We sincerely thank the reviewer for those insightful suggestions and comments, here's some of our responses.
>
> -    Part of the novelty ...
> Answer: Thanks for your suggestion. We conduct experiment using ResNet50 on ImageNet dataset. We pretrain the network to achieve accuracy of 75.2%(top-1) and 92.2%(top-5) which match the original paper. We then apply our method on all layers. It shows that our method can compress this modern architecture ResNet50 by 6.7x while maintaining top-5 accuracy(92.1%, we compare top-5 accuracy since most related work use top-5 as a standard). This result is better previous best result, including expert tuned method (2.7x [1]) and automated method (5x [2]).
> [1] Exploring the Regularity of Sparse Structure in Convolutional Neural Networks
> [2] AMC: AutoML for Model Compression and Acceleration on Mobile Devices
>
> -    Paper claims to ...
> Answer: To the best of our knowledge, [3] and many following work cannot prune all layers together. These method need to set pruning thresholds for every layer in the network, and tune those pruning thresholds carefully in order to get good performance, while our method only need one threshold for all layers. Could you please refer us some first-order methods that you’re mentioning? That would be great for us to conduct experiments and comparisons, thanks!
> [3] Learning both Weights and Connections for Efficient Neural Networks
>
> -    Paper claims little overhead...
> Answer:
> 1) Could you please refer us to which experiments you think our method has a drop-in accuracy? In our ImageNet experiment, the compressed AlexNet and VGG16 (and the updated ResNet50 experiment) both have better accuracy than the un-compressed version (Table 1 in the paper).
> 2) Our pruning operation involves computing and ranking Eq.(7), which is a little overhead compared with training and re-training. For example, in the AlexNet experiment, this time is around 70s (ResNet50 is around 40s), which is negligible compared with training(typically two days)
> 3)  We think our method is in parallel with these mentioned work. Their methods investigating different regularizer to encourage the model to have large sparsity in its parameters, and then prune unimportant ones based on the value of each parameter. Our method doesn’t investigate the effects of different regularizers, but to propose another way to select which are unimportant parameters based on Eq(7). We think their work and our work lies in two branches, and can be combined straight-forwardly, i.e. replace the l2 regularizer term in our training objective with their regularizers.
>
> -    Experiments are shown in small datasets...
> Answer: Thanks for your advice, we updated ResNet50 experiments as previously mentioned. In our original submission version, we also have experiments on large dataset such as ImageNet, please see Table 1 and section 5.1 for more details.
>
> -    Compute time is not provided...
> Answer: The major part of running time is on training and retraining (similar to other state-of-the-art pruning method). The computing time for our pruning operation is small: e.g. ~70s for AlexNet. For modern architecture which mainly consists of convolution layers, this computing time is even smaller, e.g. ~30s for ResNet. (This time is evaluated on a 32 core CPU with standard scientific libraries like Scipy and Tensorflow without further optimization) This is negligible compared to training a neural network (2~3 days)
>
> -    I am not sure ...
> Answer: As we mentioned in the introduction (paragraph 3), most of current pruning methods requires manually setting a compression threshold for each layer in the network. These thresholds need to be carefully tuned to get good result. These methods cannot directly compare the magnitudes of weights from all layers, and prune smaller ones. Our method, on the other hand, can directly compare the importance from different layers, and doesn’t require lots of per-layer compression threshold.
>
> -    Different to others...
> Answer: We think our method is a both more principled and more easier to use method. A standard or a customized modern neural network typically has 10 to 100 layers. Most previous methods needs to manually tune additional 10 to 100 hyper-parameter, and each trial involves several days of training and pruning times, which make it very time-consuming to tune and use. Our method can automatically determine those hyper-parameters, and thus can be used in a push-button-to-start manner and free from time-consuming tuning procedure. Of course, it is still possible and easy to control the sparsity of different layers: instead of ranking weights from all layers, we can ranking weights within different groups of layers, and prune separately(still using the same pruning criterion as in Eq(7)).
>
> -    I am confused with \lambda...
> Answer: \lambda is a global variable, which is equivalent to the global compression ratio in practice. We introduce it for easier illustration in formulae.

---

### Author Response · Authors · 2018-11-27
**Thank you all for reviewing our paper.**

We would like to thank all reviewers for reviewing our paper and give insightful comments. We are open to further comments.

Please find more detailed responses as below.

---

### Meta-Review · Area_Chair1 · 2018-12-14
**Work could be strengthened by analysis of runtime performance**

**Confidence:** 4
**Recommendation:** Reject

**Metareview:**

The authors propose a technique for pruning networks by using second-order information through the Hessian. The Hessian is approximated using the Fisher Information Matrix, which is itself approximated using KFAC. The paper is clearly written and easy to follow, and is evaluated on a number of systems where the authors find that the proposed method achieves good compression ratios without requiring extensive hyperparameter tuning.

The reviewers raised concerns about 1) the novelty of the work (which builds on the KFAC work of Martens and Grosse), 2) whether zeroing out individual connections as opposed to neurons will have practical runtime benefits, 3) the lack of comparisons against baselines on overall training time/complexity, 4) comparisons to work which directly prune as part of training (instead of the train-prune-finetune scheme adopted by the authors).
In the view of the AC,  4) would be an interesting comparison but was not critical to the decision. Ultimately, the decision came down to the concern of lack of novelty and whether the proposed techniques would have an impact on runtime in practice.